# Inertial Motion Capture-Based Estimation of L5/S1 Moments during Manual Materials Handling

**DOI:** 10.3390/s22176454

**Published:** 2022-08-26

**Authors:** Antoine Muller, Hakim Mecheri, Philippe Corbeil, André Plamondon, Xavier Robert-Lachaine

**Affiliations:** 1Univ Lyon, Univ Gustave Eiffel, Univ Claude Bernard Lyon 1, LBMC UMR_T 9406, F-69622 Lyon, France; 2Institut de Recherche Robert-Sauvé en Santé et en Sécurité du Travail (IRSST), Montreal, QC H3A 3C2, Canada; 3Department of Kinesiology, Université Laval, Québec, QC G1V 0A6, Canada; 4Centre Interdisciplinaire de Recherche en Réadaptation et Intégration Sociale du Centre Intégré Universitaire de Santé et de Services Sociaux de la Capitale-Nationale (CIRRIS/CIUSSS-CN), Québec, QC G1C 3S2, Canada

**Keywords:** inertial measurement units (IMU), wearable systems, workplace ergonomics, in situ analysis, kinetics, ground reaction forces

## Abstract

Inertial motion capture (IMC) has gained popularity in conducting ergonomic studies in the workplace. Because of the need to measure contact forces, most of these in situ studies are limited to a kinematic analysis, such as posture or working technique analysis. This paper aims to develop and evaluate an IMC-based approach to estimate back loading during manual material handling (MMH) tasks. During various representative workplace MMH tasks performed by nine participants, this approach was evaluated by comparing the results with the ones computed from optical motion capture and a large force platform. Root mean square errors of 21 Nm and 15 Nm were obtained for flexion and asymmetric L5/S1 moments, respectively. Excellent correlations were found between both computations on indicators based on L5/S1 peak and cumulative flexion moments, while lower correlations were found on indicators based on asymmetric moments. Since no force measurement or load kinematics measurement is needed, this study shows the potential of using only the handler’s kinematics measured by IMC to estimate kinetics variables. The assessment of workplace physical exposure, including L5/S1 moments, will allow more complete ergonomics evaluation and will improve the ecological validity compared to laboratory studies, where the situations are often simplified and standardized.

## 1. Introduction

The use of inertial motion capture (IMC) systems for measuring human movement is constantly increasing. In the context of ergonomics evaluation, IMC is used in the field for ambulatory assessment of physical workloads by analyzing human postures and movement [1]. In comparison to studies performed in the laboratory, numerous studies described the important benefits of workplace analyses [2,3,4].

Kinematic data obtained from IMC systems have been validated in several studies in the laboratory by comparison to data obtained from optoelectronic motion capture (OMC) systems, considered as a reference system. These validation studies have been performed for various applications [5,6], including work tasks with complete body movements and acquisition duration of over 30 min (Robert-Lachaine et al., 2020 [7]; Robert-Lachaine et al., 2017 [8]). The estimation of kinetic variables such as back loading from IMC systems has also been validated in the laboratory [9,10]. In this case, the use of an additional force platform was required to measure the ground reaction forces.

Because of the need to measure the contact forces between the subject and his environment (in most cases, the ground reaction forces), most in situ ergonomics studies are limited to a kinematic analysis [2,3,4]. However, the addition of physical exposure indicators based on kinetic data would allow a more complete ergonomic evaluation. Back loading is a variable commonly used for ergonomics studies [11,12,13,14,15] but rarely measured in a field. Wearable devices such as instrumented shoes or pressure insoles have been proposed to estimate ground reaction forces in the field; however, the use of instrumented shoes [16,17] influences the subject’s natural pattern [18] and force measurement errors with pressure insoles are an important limitation to its use [19].

An alternative is to estimate contact forces only from IMC kinematic data by solving the inverse dynamics problem [20,21,22]. The main limitation of this technique is that this problem is undetermined in the case of multiple contacts (double support or contact with feet and hands). Three different approaches can be used to solve this indeterminacy. First, empirical methods propose linking motion to effort [23,24], but these have only been used for walking. Second, from a database using force platform measurements, some studies have applied machine learning methods to solve the inverse dynamics problem [25,26]. The tasks studied must be standardized so that the tasks used for learning remain sufficiently similar to those studied. In addition, a large motion set is needed for the learning phase to be efficient. Third, the use of optimization techniques with a contact model has been proposed [27,28,29]. From a set of theoretical contact points, the external forces are estimated via an optimization function that minimizes the external forces or muscle forces while ensuring compliance with the equations of dynamics. In most cases, this approach is used for tasks involving foot contact only. In the case of additional hand contacts, this issue was addressed for manual material handling (MMH) tasks [30,31,32]. The use of these methods in the field is currently limited by the fact that the validation tasks were standardized and mainly carried out in a laboratory setting, so that the handling loads were carried using handles and tasks did not require moving the feet. Moreover, some motion capture systems used for validation of methods using only kinematics data, such as the OMC system, are not suitable for use in the field.

The aim of this paper was to evaluate the accuracy of an optimization-based approach using only IMC kinematics data to estimate kinetic variables, particularly back loading, during representative workplace MMH tasks. This approach was evaluated in the laboratory by comparing it to a reference method composed of an OMC system and a force platform (OMC + PF).

## 2. Materials and Methods

### 2.1. Experimental Procedure

Nine male subjects (height: 176 ± 8 cm, mass: 77 ± 14 kg) participated in the experiment. Inclusion criteria were a good physical capacity according to the Physical Activity Readiness Questionnaire (PARQ), no musculoskeletal disorder affecting work or present in the last seven days according to the Nordic questionnaire, and work experience in MMH varying between 0.5 and 5 years (2.2 ± 1.3). Exclusion criteria were a BMI over 30 and age over 60 years. The study was conducted according to the guidelines of the Declaration of Helsinki and approved by the Ethics Committee of Université Laval (QC, Canada) (2018-001 A-2, dated 20 December 2019). Informed consent was obtained from all subjects involved in the study. The tasks consisted in transferring loads from a lifting to a deposit location. The task setup was performed using a convertible experimental setup (Figure 1) that consisted of two locations used for both lifting and depositing. Four areas were available at each location: at a low level (“L”, $H_{1}=16$ cm), at a middle level (“M”, $H_{2}=116$ cm), at a high level (“H”, $H_{3}=190$ cm), and at a low level with a depth (“D”, De= 54 cm). One of the two locations was mobile, which allowed us to set up two transfer distances between the lifting and the deposit sites (Di=0.75 m and Di=1.5 m). Six different loads were used to simulate different work contexts: 3 boxes without handles (26 cm deep × 35 cm wide x 32 cm high): 2 kg, 10 kg, and 20 kg, a large 10-kg box with handles (82 × 46 × 38 cm), a 10-kg box with handles (26 × 35 × 32 cm), and a 20-kg gravel bag. For some tasks, instructions were given to the participants to impose movement variability: with or without pre-grip and post-deposit manipulations; with and without feet displacements; different lift techniques (for example, stoop or squat); free and fast pace; different controls of the load (simulation of a fragile object and release).

Four or eight repetitions were executed for each experimental condition. For two consecutive repetitions, the deposit location of the first repetition becomes the lifting location of the second repetition, and vice versa until the last repetition of the trial is executed (see Robert-Lachaine et al. [33] for more details about experimental procedures). The description of each trial is detailed in Appendix A. Overall, 156 loads were transferred by each participant.

Human motion was captured simultaneously with an IMC system (XSens MVN Link, Xsens technologies, Enschede, The Netherlands) and an OMC system (Optotrak Certus, Northern Digital Inc., Waterloo, Canada). The IMC system was composed of 17 magnetic and inertial measurement units (MIMU) (head, scapulae, sternum, pelvis, arms, forearms, hands, thighs, shanks, and feet), sampled at 240 Hz (Figure 2). For the OMC system, a three-marker cluster was rigidly fixed on each MIMU (Figure 2). Eight cameras recorded at 40 Hz the 3-D coordinates of these markers. At the beginning of each experiment, the locations of 45 anatomical landmarks were identified in relation to their respective cluster. The ground reaction forces (GRF) were measured by a large homemade force platform (1.90 × 1.40 m) at 1000 Hz [34].

All the trials were captured using three video cameras (20 Hz) to identify the transfer phases (see Section 2.2). The OMC system, force platform, and video cameras were hardware synchronized. The OMC and IMC systems were synchronized by using a dedicated high-speed flexion/extension elbow movement before each trial, where the peaks of hand velocity signals were used.

### 2.2. Transfer Phase Identification

The transfer phase corresponded to the period of time when the participant fully supported the load, which is the most relevant phase for ergonomics studies. The start and end instants of each transfer phase were manually identified from the video cameras. The start instant has been identified as the instant when the load is no longer in contact with the lifting support. The end instant has been identified as the instant when the load initializes its contact with the deposit support. Only the transfer phases were considered in this study.

### 2.3. OMC + PF-Based Computation

The biomechanical model used was composed of 18 rigid segments (head, upper trunk, lower trunk, pelvis, clavicles, arms, forearms, hands, thighs, shanks, and feet) linked by 17 joints corresponding to 43 degrees of freedom. For each joint, the sequences of rotation were chosen to follow the ISB recommendations [35,36]. In particular, for the L5/S1 joint, the successive rotation angles corresponded to flexion/extension (*z*-axis), lateral bending (*x*-axis), and axial rotation (*y*-axis). The geometrical parameters were subject-specific calibrated using motion capture data and an optimization-based method [37]. Body segment inertial parameters were estimated from an anthropometric table [38]. From the positions of the 45 anatomical landmarks (estimated from the locations of the 12 rigid clusters) the joint coordinates were computed using a multibody kinematics optimization approach [39].

An inverse dynamics step was then performed to obtain intersegmental joint moments. The measured GRF was used in a recursive Newton–Euler algorithm with a bottom-up approach. The velocities and the accelerations were computed with a second-order finite-difference method. To improve this computation, the joint coordinates were beforehand filtered with a fourth-order Butterworth low pass filter with a cut-off frequency of 6 Hz and no phase shift.

### 2.4. IMC-Based Computation

Data from MIMU were processed with MVN Analyze software [40] which integrates a sensor fusion algorithm. The joint coordinates and the biomechanical model were extracted from this software by using an *mvnx* file. The MVN biomechanical model (XSens built-in) is composed of 23 rigid segments (pelvis, L5, L3, T12, T8, neck, head, shoulders, upper arms, lower arms, hands, upper legs, lower legs, feet, and toes) where each joint is assimilated to a spherical joint. The same anthropometric table used for the OMC + PF-based computation approach was used to compute the body segment inertial parameters. Five new segments (head + neck; T8 + T12 + shoulders; L5 + L3; left foot + toe; right foot + toe) were formed by the combination of MVN model segments to match the segment definitions of the anthropometric table [23].

The external forces estimation approach was the same as the one previously detailed [31,32]. This approach used discrete contact points under feet and on hands: 8 points under each foot and 8 points on each hand [31,32]. These points were located in order to map the contact area of the biomechanical model. At each sample over time, a contact point was considered active if an external force could be applied on this point. A contact point under a foot was active if it was sufficiently close to the ground at a sufficiently low speed [41,42]. As only transfer phases were analyzed, the contact points on the hands were always considered active. Contact forces were estimated using an optimization procedure, which consisted of minimizing the sum of squared contact forces respecting the dynamic equations applied on the subject and applied on the load. The center of mass of the load was estimated in the middle of the subject’s hands and only its mass was considered. Then, using the estimated GRF, the same inverse dynamics step as for OMC + PF-based computation (see Section 2.3) was applied.

### 2.5. Data Analysis

OMC + PF and IMC-based computation were compared in terms of GRF, the center of pressure (CoP) position, and L5/S1 moments. As the expression of the coordinate systems of both computations had only the vertical axis in common, the resultant force of the anteroposterior and mediolateral axes (transverse force) was used for comparison. As the origin of the two coordinate systems was not the same, the relative position of the CoP is expressed in terms of the pelvic coordinate system located at the L5/S1 joint. The position analyzed is that of its projection in the horizontal plane. Considering all trials of all participants, root mean square error (RMSE) of vertical and transverse GRF, antero-posterior (AP) and medio-lateral (ML) components of the relative position of the CoP, and flexion and asymmetric (combination of the lateral bending and axial rotation components) L5/S1 moments were computed. RMSE was also calculated for each load mass: 2 kg, 10 kg, and 20 kg.

Moreover, L5/S1 peak and cumulative moments were evaluated for each transfer with both computations and then compared. These measures are typical indicators used in ergonomics studies [33,43,44]. For each transfer, the flexion and asymmetric L5/S1 peak moment were identified for the OMC + PF-based computation. To make sure to identify the IMC-based computation peak relative to the same phase of the transfer as for the peak of OMC + PF, this peak was identified in a range of 100 frames (at 240 Hz) centered on the identified instant peak of the OMC + PF. Bland-Altman bias (b), confidence interval (CI; 1.96 times standard deviation or 1.45 times interquartile range for non-normal distributions) as well as the coefficient of determination (R^2^) and RMSE were calculated.

## 3. Results

RMSE between GRF, the relative position of the CoP, and L5/S1 moments computed with both OMC + PF and IMC are presented in Table 1. The errors on GRF were higher for the vertical force than for the transverse. The errors on the relation position of the CoP were, on average, 3.5 cm along AP and ML directions. No visible difference was observed between these two directions. Concerning back loading assessment, the errors on L5/S1 flexion moments were higher than for asymmetric moments. For each of these variables, the increase in handled mass is associated with an increase in the estimation error.

Correlations between OMC + PF-based and IMC-based computations of L5/S1 peak moments are displayed in Figure 3 and corresponding Bland–Altman plots are displayed in Appendix A. For the flexion component, the variety of MMH tasks performed resulted in peak values ranging from 12 Nm to 340 Nm for the OMC + PF-based computation, with an average of 174 Nm. For the asymmetric component, the peak values were between 14 Nm and 160 Nm for the OMC + PF-based computation, with an average of 58 Nm. The RMSE is quite similar for both components (Table 2); the coefficient of determination was higher for the L5/S1 peak moments. For both components, a bias of about −17 Nm was observed, which means that the IMC-based computation generally underestimates the OMC + PF value.

Correlations between OMC + PF-based and IMC-based computations of L5/S1 cumulative moments are displayed in Figure 4, and corresponded Bland–Altman plots are displayed in Appendix A. For the flexion component, the cumulative values were between 1 Nm·s and 640 Nm·s for the OMC + PF-based computation, with an average of 172 Nm·s. For the asymmetric component, the cumulative values were between 4 Nm·s and 195 Nm·s for the OMC + PF-based computation, with an average of 50 Nm·s. RMSE errors were found higher for the flexion component with a coefficient of determination above 0.95. Biases of about −15 Nm·s were observed for flexion and asymmetric components, which means that the IMC-based computation generally underestimates the OMC + PF value.

## 4. Discussion

The present study evaluated an IMC-based computation approach for estimating kinetic variables, especially back loading, during continuous sequential lifting, carrying, and lowering. GRF and L5/S1 moments estimated from IMC were compared to an OMC + PF-based computation. On a large variety of representative workplace MMH tasks, acceptable errors and excellent correlations were found on L5/S1 moments estimation for the flexion component, while lower correlations were found for the asymmetric component.

### 4.1. IMC-Based Estimation Accuracy

The external forces estimation method is based on dynamic equations. Most errors come from kinematics [8,9] and inertial parameters of body segments [20]. It has been reported that the part of errors due to the kinematics is preponderant in the dynamic equations and especially in the calculation of dynamic residuals [45]. This is probably accentuated using an IMC system where, for example, the need for a calibration phase and drift phenomena induces errors in the kinematics. Moreover, the estimation of the contact forces distribution is a critical challenge in the case of multiple contacts between the subject and the environment [20] since an infinite number of solutions respecting the dynamic equations exist. The assumption of selecting the solution minimizing a cost function necessarily induces additional errors.

The errors obtained on the GRF estimation (mean of 39.6 N for vertical force and mean of 24.8 N for the transverse force) are of the same order of magnitude or lower than similar methods developed in the literature. By using the XSens IMC system, Larsen and Svenningsen [46] reported estimation errors of 89 N, 17 N, and 39 N on vertical, anterior-posterior, and mediolateral GRF components, respectively. By using the Perception Neuron IMC system, Diraneyya et al. [47] reported estimation errors of 84 N, 61 N, and 88 N on vertical, anterior–posterior, and mediolateral GRF components, respectively. For this latter study, these larger errors are probably due to kinematics errors provided by the low-cost IMC system [7]. On L5/S1 moments estimation, the errors obtained in this current study (mean of 21.4 Nm for the flexion component and mean of 15.6 Nm for the asymmetric component) are slightly higher than those reported for trunk bending tasks (between 5 Nm and 10 Nm on each axis [21]) or for MMH tasks measured with an OMC system (14 Nm for flexion and about 10 Nm for asymmetric component [31]). The latter study also considered phases where there is no effort applied to the hands. For both GRF and L5/S1 moments estimation, Delisle et al. [48] reported errors of the same order of magnitude using an OMC system and considering only the resultant GRF. Thus, considering MMH tasks representative of the workplace, the errors obtained here seem acceptable.

Several indicators used in ergonomics studies are based on back loading that can be characterized by the L5/S1 moments. The use of an IMC-based computation will introduce some estimation errors compared to an OMC + PF-based computation, considered as a reference for this type of analysis. On all the subjects and tasks studied, excellent correlations between both computations were obtained for indicators based on flexion L5/S1 moments. However, lower correlations were obtained for indicators based on asymmetric L5/S1 moments, particularly for the peak values. Since the error on the relative position of the CoP was of the same order of magnitude in the AP and ML direction, this error had a similar impact on both components of L5/S1 moments. The lower amplitude for the asymmetric moment thus led to lower correlations.

In addition, the mass of the load seems to have a significant effect on the estimation errors: the higher the mass, the larger the error. The heavier the load, the more the kinematic errors will have an important impact on the kinetics. In particular, errors in estimating the position of the hands in relation to the L5/S1 joint can explain this phenomenon. The higher the mass of the load, the higher the error in the estimation of the moment arm will influence the L5/S1 moment estimation. With the use of an OMC system, the moment arm is directly evaluated from the position of markers. With an IMC system, this moment arm is indirectly estimated from the orientation of several segments and by using a multi-segmental model. Thus, for example, a small error in the estimation of the shoulder kinematics or a small error in a limb length can lead to a large error in the relative position of the hands. Many studies have evaluated the accuracy of IMC systems in terms of joint angles, but few on the estimation of the relative position of anatomical landmarks such as the moment arm between the L5/S1 joint and the hands. Koopman et al. [10] reported an error on this moment arm, at the instant of peak moment, on average 10 ± 4 cm. This error corresponds to the order of magnitude of the error induced on the L5/S1 moments estimation by the increase in mass of the load. In addition, the load was modeled here as a point mass, whose kinematics was estimated from the kinematics of the hands. The approach used has an influence on the dynamics [49]. The influence of this simplified modeling increases when the mass of the load increases.

### 4.2. Prospects for Use

The results obtained showed excellent correlations between the estimation of flexion L5/S1 peak and cumulative moments. This study shows the potential of using only IMC to estimate these indicators in ergonomics studies in a wide range of MMH tasks.

Back loading estimation based on IMC opens up perspectives for in situ applications; for example, to evaluate working situations [33,43], evaluate MMH techniques, or study the influence of individual characteristics [11,50]. Conducting these studies in the field will improve the ecological validity compared to laboratory studies, where the situations are often simplified and standardized. The computation method developed in this study is based only on the full body kinematics of the handler, measured with an IMC system, and some additional information: mass of the load and instants of the transfer phase. Apart from the MIMU placed on the body segments, no force measurement (GRF or load contact forces) and no measurement of the load kinematics are needed. This avoids the need for instrumentation of the working environment and keeps the environment as it is naturally. The use of indicators based on asymmetrical L5/S1 moments should be used with precaution since the correlations obtained between both computations were lower. The identification of the main sources of error for the evaluation of these indicators is an important future step. For errors coming from kinematics, few studies focus on variables such as the moment arm between the L5/S1 joint and the hands or segmental accelerations. A correct evaluation of these variables is, however, essential for this type of motion-based estimation method. Moreover, a study focusing on the cost function used in the optimization procedure could improve the results. This cost function should enable to mimic as well as possible the distribution function of the efforts between the different contact points. As this function is task-dependent, a particular study is necessary in the case of MMH tasks. Considering internal forces, for example intersegmental joint moments, could be an avenue for improvement.

The identification of the transfer phases was here made manually by video observations, which can be time-consuming. The development of automatic identification would facilitate the use of the IMC-based method in the workplace. A machine learning method based on hand kinematics has been proposed in the literature [32] but needs to be validated in MMH tasks such as those performed in this study.

A wide variety of handling tasks was considered in this study. The conditions and instructions imposed allowed for variations in the type of foot displacements (no movement vs. several steps), in handled loads (masses, with or without handles, stiffness), in dynamics (slow vs. fast), in techniques (low vs. high, squat vs. stoop, torsion or not), etc. These tasks were intentionally designed to cover multiple variations of working contexts and individual performance of MMH tasks can be individually performed, which should be representative of many occupational settings.

### 4.3. Limitations

This study presents several limitations. First, both computation methods (IMC and OMC + PF) used their own biomechanical model. The reported differences come from different sources: kinematics, biomechanical model, and external contact forces. The comparison performed does not allow to distinguish the influence of these different sources. Second, although the elaborated method estimates the forces applied on each foot and each hand, only the resultant GRF was evaluated because it was measured by one large force platform. Several force platforms or hand force sensors have been used in other similar studies [17,30,32,51] but have the disadvantage of limiting footstep strategies or requiring loads with instrumented handles. The use of this IMC-based method for the computation of lower limbs or upper limbs intersegmental moments would require further validation. Third, only a relatively small group of male subjects participated in this study. A similar study with female participants would be relevant to provide a more complete validation. Finally, the experiments were performed in a laboratory context with controlled magnetic disturbances. The use of IMC systems being sensitive to these disturbances [52], a feasibility study in the workplace, comparable to what has been performed for kinematics [53], would be interesting.

## 5. Conclusions

The current study proposed and evaluated an IMC-based computation approach for estimating L5/S1 moments during MMH tasks. Various tasks and working contexts were analyzed to be representative of many occupational settings. For all these tasks, acceptable levels of error were found on GRF estimation and L5/S1 moments. Excellent correlations were obtained for ergonomic indicators based on flexion L5/S1 peak and cumulative moments between IMC and OMC + PF computations. Lower correlations were obtained for indicators based on asymmetric L5/S1 moments, particularly for the peak values. This approach permits us to estimate these indicators only based on the handler’s kinematics measured by an IMC system, the mass of the load, and the instants of the transfer phase. This avoids the need for instrumentation of the working environment and keeps the environment as it is naturally. The addition of physical exposure indicators based on kinetics, such as back loading, will allow more comprehensive assessments directly in the workplace, for example, to evaluate working situations, evaluate MMH techniques, or study the influence of individual characteristics.

## Figures and Tables

**Figure 1 sensors-22-06454-f001:**
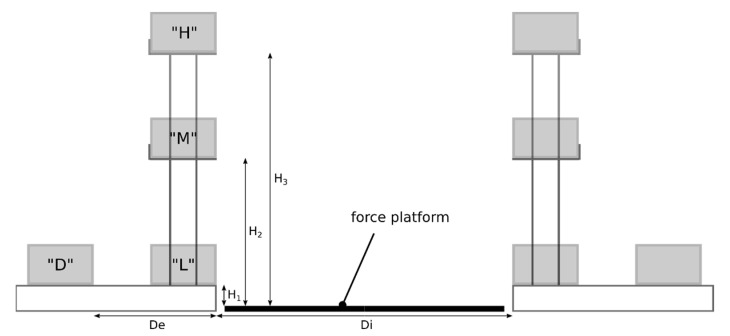
Experimental convertible setup consisting of two areas used for lifting and deposit locations. “D”, “L”, “M”, and “H” indicate the four areas available for both lifting and deposit. $H_{1}$, $H_{2}$, and $H_{3}$ indicate the low, middle, and high level of the load, respectively; $D_{e}$ represents the distance of a potential depth; and $D_{i}$ represents the distance between the lifting and the deposit sites.

**Figure 2 sensors-22-06454-f002:**
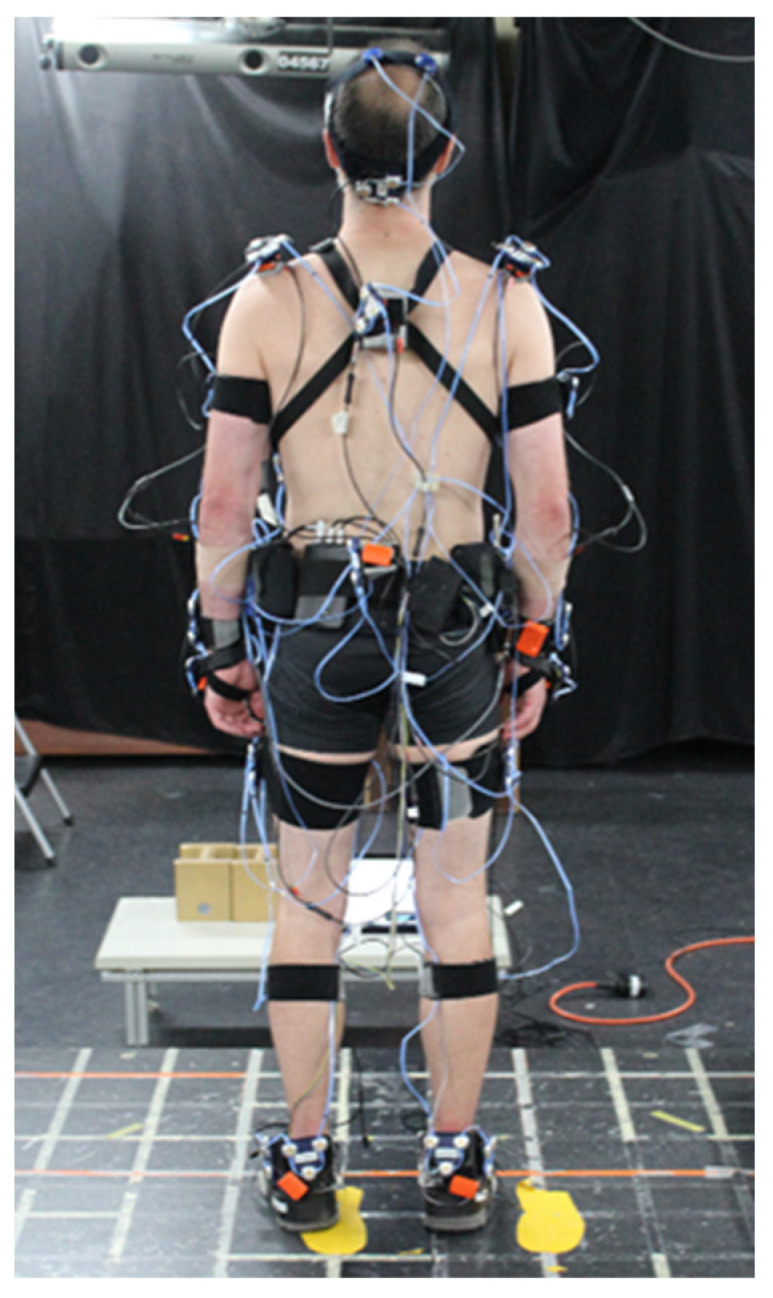
Experimental setup with the IMC system, the OMC system, and the force platform.

**Figure 3 sensors-22-06454-f003:**
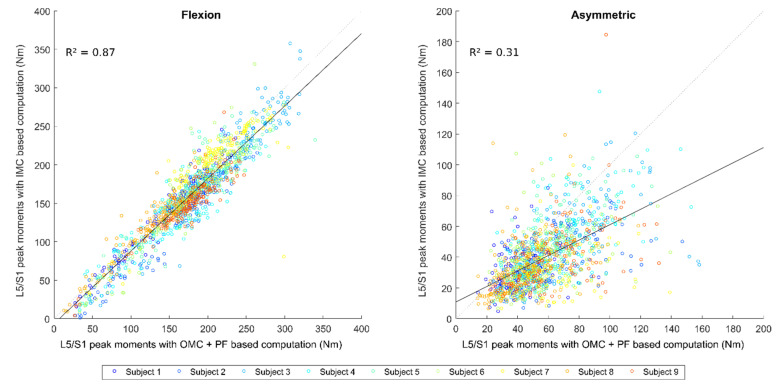
Correlation plot between OMC + PF-based and IMC-based computations of L5/S1 peak moments.

**Figure 4 sensors-22-06454-f004:**
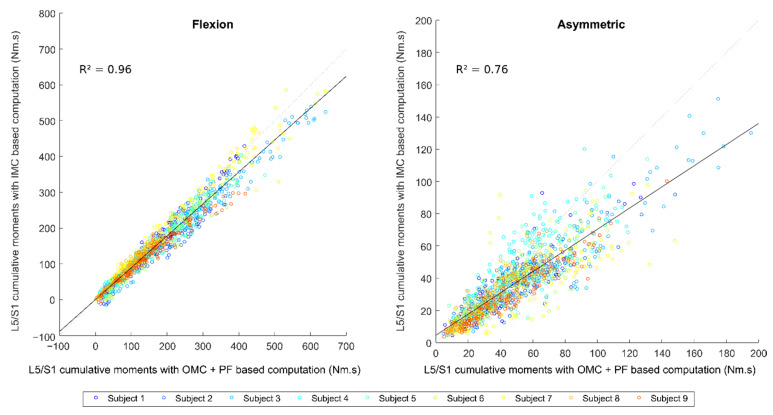
Correlation plot between OMC + PF-based and IMC-based computations of L5/S1 cumulative moments.

**Table 1 sensors-22-06454-t001:** RMSE between OMC + PF and IMC for the GRF, relative position of the CoP, and L5/S1 moments curves during the transfer phase.

RMSE	Ground Reaction Force (GRF)		Relative Position of the Center of Pressure (CoP)		L5/S1 Moments	
	Vertical	Transverse	AP	ML	Flexion	Asymmetric
All	39.6 N	24.8 N	3.5 cm	3.5 cm	21.4 Nm	15.6 Nm
2 kg	27.1 N	19.2 N	3.0 cm	3.3 cm	15.1 Nm	12.0 Nm
10 kg	40.0 N	24.9 N	3.5 cm	3.6 cm	21.4 Nm	15.8 Nm
20 kg	52.9 N	30.9 N	4.0 cm	3.6 cm	28.5 Nm	19.1 Nm

**Table 2 sensors-22-06454-t002:** Bland–Altman bias (b), confidence interval (CI), as well as the coefficient of determination (R^2^) and RMSE between OMC + PF-based and IMC-based computations for L5/S1 peak and cumulative moments.

		RMSE	b	CI	R^2^
Peak	Flexion	26.5 Nm	−16.4 Nm	35.0 Nm	0.87
	Asymmetric	27.0 Nm	−17.8 Nm	30.1 Nm	0.31
Cumulative	Flexion	31.6 Nm·s	−18.3 Nm·s	41.8 Nm·s	0.96
	Asymmetric	18.6 Nm·s	−12.1 Nm·s	22.9 Nm·s	0.76

## Data Availability

Not applicable.

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
