# Peer review of "Inertial Motion Capture-Based Estimation of L5/S1 Moments during Manual Materials Handling"

_sensors, 2022, doi:10.3390/s22176454_

Round 1

Reviewer 1 Report

Dear Authors

The manuscript, submitted for review, is interesting and the topics covered-current and necessary in a work environment where proper ergonomics is valued and cared for. I have some reservations and comments about the content and I ask the authors to respond. In the Abstract section, please explain to readers the concept of kinematic analysis. Please also explain what specifically the authors mean when they write about kinetic variables? In the Introduction, please correct the cited literature according to the requirements of the journal. In Materials and Methods, please explain why only men took part in the experiment? What do the authors mean when they write healthy males? Perhaps it would be appropriate to rephrase this sentence, since health is a very broad concept, not only in medicine? The experiment was conducted on a small group. I think it would be a valuable addition to analyze a similar group of women, the experiment would gain more reliability and correctness. The results presented are interesting, including the discussion, so it would be good to complete the Conclusions section based on them.

Regards

Reviewer 2 Report

In the submitted manuscript, the authors developed an inertial motion capture (IMC) approach to estimate kinetic quantities relevant for ergonomic evaluations, particularly as it relates to lower back loading. Nine participants completed a range on manual material handling tasks resulting in 156 trials per participant. Inverse dynamics was conducted with optical motion capture and force plate data following a multibody kinematics optimization approach, which was used as the reference for comparison. The IMC-based kinetics results leveraged the kinematic outputs from Xsens proprietary software in an optimization procedure that found contact force solutions that adhered to the dynamics exhibited by the system. For each load mass, metrics used for evaluation included ground reaction forces (vertical and transverse), center of pressure positions (antero-posterior and medio-lateral), and L5/S1 moments (flexion and asymmetric).

One major/minor revisions that should be addressed prior to publication relates to the description of the IMC-based computation (Section 2.4) is not complete enough that a reader would be able to recreate the approach. The aim of the paper is to evaluate the accuracy of this approach, but it is not actually fully described. As a result, it is unclear how this work differs from the work/validation that was been published previously (i.e., [31-32]).

A few minor revisions are also listed below:

·        The phrase “Error! Reference source not found” appears multiple times in the manuscript (e.g., lines 84, 112, 113/114, 198, 209, 214, and 222).

·        Please define “MMH” in the abstract before using the acronym.

·        On line 173, Section “0” is referenced. Also, “OMC + PF” instead of “OMC + FP”.

·        Did the authors observe any heteroscedasticity in the Bland-Altman plots? If so, could they provide the plots themselves in a supplemental document?

Overall, this manuscript was well-written and communicated. It is the opinion of this reviewer that the manuscript be accepted after major/minor revisions.
